# Virtual Reality-Based Psychoeducation for Dementia Caregivers: The Link between Caregivers’ Characteristics and Their Sense of Presence

**DOI:** 10.3390/brainsci14090852

**Published:** 2024-08-23

**Authors:** Francesca Morganti, Maria Gattuso, Claudio Singh Solorzano, Cristina Bonomini, Sandra Rosini, Clarissa Ferrari, Michela Pievani, Cristina Festari

**Affiliations:** 1Department of Human and Social Sciences, University of Bergamo, 24129 Bergamo, Italy; maria.gattuso@unibg.it; 2CHL—Centre for Healthy Longevity, University of Bergamo, 24129 Bergamo, Italy; 3Laboratory of Alzheimer’s Neuroimaging and Epidemiology, IRCCS—Istituto Centro San Giovanni di Dio Fatebenefratelli, 25125 Brescia, Italy; csolorzano@fatebenefratelli.eu (C.S.S.); mpievani@fatebenefratelli.eu (M.P.);; 4U.O. Alzheimer, IRCCS—Istituto Centro San Giovanni di Dio Fatebenefratelli, 25125 Brescia, Italy; 5Research and Clinical Trials Office, Istituto Ospedaliero Fondazione Poliambulanza, 25124 Brescia, Italy

**Keywords:** virtual reality, dementia, presence, psychoeducation, empathy, distress, caregiver burden

## Abstract

In neuropsychology and clinical psychology, the efficacy of virtual reality (VR) experiences for knowledge acquisition and the potential for modifying conduct are well documented. Consequently, the scope of VR experiences for educational purposes has expanded in the health field in recent years. In this study, we sought to assess the effectiveness of ViveDe in a psychoeducational caregiver program. ViveDe is a VR application that presents users with possible daily life situations from the perspective of individuals with dementia. These situations can be experienced in immersive mode through 360° video. This research aimed to ascertain the associations between the sense of presence that can be achieved in VR and some users’ psychological characteristics, such as distress and empathetic disposition. The study involved 36 informal caregivers of individuals with Alzheimer’s disease. These participants were assessed using scales of anxiety and depression, perceived stress, empathy, and emotional regulation. They were asked to participate in a six-session psychoeducation program conducted online on dementia topics, in addition to experiencing the ViveDe application. The immersive VR sessions enabled the caregivers to directly experience the symptoms of dementia (e.g., spatial disorientation, agnosia, difficulty in problem-solving, and anomia) in everyday and social settings. The results indicated that although the experience in ViveDe (evaluated using the XRPS scale and five questions about emotional attunement) showed efficacy in producing a sense of first-person participation in the symptoms of dementia, further research is needed to confirm this. The structural equation model provided evidence that the characteristics of individuals who enjoy the VR experience play a determining role in the perceived sense of presence, which in turn affects the efficacy of the VR experience as a psychoeducational tool. Further research will be conducted to ascertain the potential role of these elements in conveying change in the caregivers of people with dementia. This will help us study the long-term effectiveness of a large-scale psychoeducation program in VR.

## 1. Introduction

Upon diagnosis of dementia within a family, an informal caregiver is identified. This is the individual, often a female with a very close relationship with the person with dementia, who assumes responsibility for meeting their needs despite lacking the specific qualifications to do so. The most striking aspect of this transformation in family roles is the pervasive belief that the newly designated caregiver is fully capable of meeting the needs of the person with dementia. This includes having an appropriate attitude toward the individual and, most crucially, the emotional resilience to cope with the challenges of caregiving [1]. In the majority of cases, however, informal caregiving has been found to increase the risk of depression and anxiety, particularly in women and caregivers with limited experience [2,3]). This situation frequently results in the emergence of what is referred to as the caregiver burden. This is a situation in which, as a consequence of the inability to manage the emotional and cognitive burden experienced by the caregiver, the individual with dementia is initiated into institutionalization at an early stage [4]. In light of this, it is evident that a considerable amount of recent research is being conducted with the aim of elucidating the factors that contribute to the adverse psychological effects of informal caregiving for individuals with dementia. This information is crucial for the development of interventions designed to support the caregiving relationship. However, to date, the effectiveness of these interventions remains moderate [5,6].

To improve the situation, psychoeducational projects have been developed in recent years with the aim of providing the family members of people with dementia with a greater understanding of the symptoms associated with the ills they are suffering [7]. Psychoeducation programs are regarded as interventions designed to equip informal caregivers with the knowledge and skills necessary to fulfill their role effectively. The objective of these programs is to address the deficit in knowledge regarding the progression of dementia and the management of associated symptoms. They seek to equip family members with the requisite skills to provide daily care and to orient themselves to the needs of caregivers and the provision of emotional support [8].

Most of these programs have been demonstrated to be quite beneficial in assisting caregivers in managing symptoms and disease progression [9,10]. To illustrate, a recent meta-analysis conducted by Gosh and colleagues [11] examined the efficacy of 24 short-term and long-term period psychoeducational interventions for dementia. The interventions included information sessions on dementia and support services, behavioral training, coping strategies, cognitive and behavioral therapy, problem solving, and counseling. The included studies appear to be effective in reducing distress and emotional disruption, as well as having moderate or heterogeneous effects in reducing anxiety and depression and improving quality of life. However, substantial between-study variance was observed, and several studies showed no significant improvement in caregiving well-being, communication with patients, or coping with the disease.

However, few psychoeducational programs have been designed with the specific objective of accompanying caregivers in understanding the disease perspective from the patient’s point of view, with the aim of facilitating a more mindful and empathic approach to care [12,13].

The objective of the investigation we present here is to identify the resources a caregiver can rely on to build a caring relationship with a person with dementia. Among these resources, the ability to take on the other person’s point of view and to come to a state of emotional attunement with the person seeking care has been highlighted [14]. Additionally, the importance of empathy and the ability to recognize and respond to emotional cues was emphasized in studies conducted by Shim, Barroso, and Davis [15], and Sutter et al. [16], respectively. This type of perspective shift is an imaginative process in which, as an observer of a cognitive and emotional change that they cannot experience firsthand, the caregiver must actively strive to take on an “other-than-self” perspective when assessing the care needs of the person with dementia without being able to truly understand them fully [17,18]). Perspective-taking is of particular importance in dementia caregiving, because the primary objective of the caregiver is to provide high-quality care while reducing the burden of care for themselves. However, if the perspectives of the caregiver and the care receiver do not align, this discrepancy may result in continued and unwarranted frustration for the caregiver [19]. This frustration has two adverse consequences. First, it increases the stress level of the caregiver. Second, it frequently results in the selection of inappropriate or unnecessary care options for people with dementia.

Precisely with the aim of guiding caregivers toward a better understanding of the disease symptoms and the feelings experienced by people with dementia to support an empathic attunement with them, educational–experiential proposals have emerged in recent years that use immersive technologies (usable through virtual reality [VR] devices) capable of making the viewer assume “the point of view of a person with dementia” [20,21]. In stark contrast to the numerous VR applications that have been introduced in recent decades for the purposes of neuropsychological assessment and symptom treatment [22], this distinct approach to the utilization of VR enables the identification and utilization of the sense of belonging to the virtual experience as a pivotal element in the generation of tangible change in the user. Indeed, the scientific literature indicates that one of the primary effects of VR is the generation of a sense of subjective involvement, defined as “presence”, through immersion, storytelling, and activity within a virtual environment. In essence, the concept of presence can be described as the subjective experience of being situated within a virtual environment [23,24]). While the characteristics of virtual reality technology can facilitate varying degrees of immersion in a virtual environment, the sense of presence experienced by participants is the result of a complex interplay between human, contextual, and virtual environmental factors. It is the precise sense of presence and belonging to the environment being explored that elicits in the user a series of first-person reactions that allow them to gain new knowledge and apply it in the real world [25].

Consequently, for those in a caregiving capacity, having the ability to more deeply understand symptoms and to empathize with a person with dementia through a 3D VR experience can assist them in a more comprehensive understanding of the needs and expectations of their care recipients.

Additionally, the experience may prompt reflection on the motivations and underlying needs of individuals with dementia who engage in certain behaviors, including “running away from home”, hiding objects, and refusing food.

In such a scenario, family members are better able to refrain from interpreting certain aberrant behaviors as intentional defiance of their role as caregivers and instead recognize that these behavioral abnormalities may result from the illness that leads the person with dementia to engage in behaviors that are often perceived as problematic in the relationship.

Similarly, increasing this kind of awareness through the sense of presence generated in VR situations may result in the caregiver experiencing a reduction in stress levels or a decrease in the experience of what is known as the caregiver burden.

Virtual reality underwent significant advancement towards the end of the 20th century and throughout the early 2000s, demonstrating its efficacy as a conduit for knowledge acquisition. As an innovative technology, however, it is not without limitations. One such limitation is the cost of the technology, which has fortunately decreased significantly in recent years, thereby enabling the deployment of VR even on limited budgets. A further significant challenge associated with the use of experiential technologies, such as VR, is the individual sensitivity to cybersickness and the potential for discomfort experienced by some users when interacting with immersive VR environments. Notwithstanding these constraints, virtual reality (VR) continues to represent a valuable instrument for enabling individuals to directly experience circumstances that are not feasible in actual life but can be simulated in a virtual environment.

In any case, given the diversity of disease pathways that dementia can have and the many differences in the roles and commitments that caregivers have in the care journey, it can be postulated that the effectiveness of the virtual experience may not be equivalent for everyone. Moreover, precisely because it is subjective, in this research we seek to identify whether the sense of presence generated in immersion with a VR environment is equivalent for every caregiver and whether it depends on the characteristics of the virtual environment itself or the individual characteristics of the caregivers themselves.

Consequently, we seek to ascertain the relationship between presence and immersion in VR in a cohort of informal caregivers of individuals with Alzheimer’s disease (AD). To ascertain whether the characteristics of the caregivers and care receivers may have influenced the sense of presence experienced in VR, and thus the effectiveness of a psychoeducation program using immersive technologies, we took into account the aforementioned characteristics.

## 2. Methods

### 2.1. Participants

A total of 36 informal caregivers of community-dwelling individuals with mild to moderate AD were recruited at the IRCCS Fatebenefratelli Memory Clinic in Brescia (Italy) among their AD-assessed patients. The mean age of the participants was 53.81 ± 10.81 years, and 80.6% of the participants identified as female. The participants were all self-defined as the primary caregivers for a person with dementia for at least 4 h per day over the previous 6 months prior to enrollment. Individuals who had previously participated in dementia educational programs, were currently undergoing psychotherapy, or required psychological assistance for caregivers focused on role difficulties were excluded from the study. In addition, participants were excluded if they lacked access to an internet-connected device (e.g., smartphone or PC) or if they had contraindications for VR use.

To qualify, participants had to be caregivers of people who had received a diagnosis of mild Alzheimer’s disease, as indicated by a score on the Mini-Mental State Examination falling between 18 and 24. The mean age of the AD patients was 77.08 ± 2.23 years, and the mean duration of the disease was 2.19 ± 1.69 years.

This study was carried out in accordance with the principles outlined in the Declaration of Helsinki, and all participants provided written informed consent.

The study was approved by the ethics committee of the IRCCS Centro San Giovanni di Dio–Fatebenefratelli in Brescia, Italy (approval date: 8 April 2022; approval number: 19/2022).

### 2.2. Measurements

Prior to their involvement in the psychoeducation program, the participants underwent an assessment to ascertain their levels of anxiety, depression, and perceived stress, as well as their skills in empathy and emotional attunement. The evaluation involved the administration of standardized clinical scales that have shown construct validity in the scientific publications below. As for the in-depth study of perceived sense of presence, a short survey constructed specifically for this research was added to the standardized rating scale. All scales were either self-administered or proposed by a psychologist.

To measure state and trait anxiety, the State–Trait Anxiety Inventory (STAI-Y, 1&2; [26]) was utilized. Depressive symptoms were assessed using the Beck Depression Inventory (BDI-II; [27]). Caregiver burden was assessed the Zarit Burden Inventory (ZBI; [28]). To assess empathetic disposition and emotion regulation strategies, the Interpersonal Reactivity Index (IRI; [29]) and the Emotion Regulation Questionnaire (ERQ; [30]) were employed. The sense of presence generated from the VR experience was tested in each meeting using an Italian-adapted version of the Extended Reality Presence Scale (XRPS; [31]) and an ad hoc emotional attunement questionnaire, which the caregivers were asked to complete immediately after each of the 360° ViVeDe videos.

Additionally, the caregivers were asked to provide information about their sociodemographic characteristics and the clinical features of the person with dementia (including age, level of education, Mini Mental State Examination score, and duration of illness). This information was collected by a psychologist who had assessed the psychic and behavioral symptoms of the patients using the Neuropsychiatric Inventory (NPI; [32]).

### 2.3. Materials

To facilitate the direct experience of the psychoeducational content conveyed by the meetings, a 10 min session was offered for the viewing of immersive 360° video content. These videos, designed, developed, and optimized for psychoeducational use, form an integral part of the ViveDe project (www.vivede.it; [20]).

ViveDe, a project developed at the University of Bergamo, employs immersive VR technologies to enable viewers to adopt the perspective of an individual with dementia. The 360° videos featured in ViveDe were were developed through a comprehensive analysis of the characteristics associated with dementia, drawing upon scientific literature and a field observation of behaviors enacted by people with dementia in everyday settings, as well as focus groups conducted with caregivers of people with dementia that highlighted certain recurring patterns of activity of their care receivers. It was originally developed with the aim of fostering a dementia-friendly and inclusive culture for individuals with dementia. It is freely available online (https://www.youtube.com/playlist?list=PLHkLQH-uZrEirVL6-sX24K8pqiSyhgV-D; accessed on 11 June 2024). ViveDe consists of 360° videos that can be explored in VR. These videos depict various everyday scenarios, including both indoor and outdoor settings, that enable users to experience specific symptoms of dementia, such as disorientation, agnosia, apraxia, and memory loss. Additionally, the videos depict the challenges that a person with dementia faces on a daily basis.

The ViveDe videos were created with the intention of being utilized in conjunction with a commercial VR device for smartphones (e.g., Headset Cardboard V2). They have been optimized for lenses with a field of view of 100°, presbyopia and myopia settings, and IPD and immersion settings. The audio is audible when played through headphones. In contrast to the two-dimensional videos, the 360° videos permit the exploration of a scene from a first-person perspective from all directions. This enables participants to perceive the entire scene as if they were a person with dementia. In addition, the audio can be adjusted to match the participant’s own voices, allowing them to hear the content as if it were their own thoughts.

A series of hypothetical scenarios were presented to the participants, each designed to elicit their personal experiences with a particular issue related to dementia care. These included the following:The experience of receiving a diagnosis of dementia with a lack of communication between the doctor and the patient;The experience of difficulties that the patient might have in preparing a meal at home;The experience of spatial and temporal disorientation in a public place and when meeting unknown people;The experience of the difficulties the person may have in purchasing daily groceries;The experience of unfamiliarity when returning to their own home;The experience of the difficulties a person might face at a family meal.

### 2.4. Procedure

The role of presence in the VR-based psychoeducational intervention was observed in a larger study designed as a two-arm randomized clinical trial that included (i) a psychoeducational intervention and (ii) a psychoeducational intervention combined with VR ([33]; clinicaltrial.gov NCT05780476). Both interventions were delivered by experienced psychologists in an online video session. It was provided in a group session with a maximum of 10 caregivers at a time and took place in weekly sessions of 1 h each for six sessions. Meeting topics included an overview of the progression of Alzheimer’s disease, recognition and management of cognitive symptoms, environmental adjustment, and stress management ([34]).

In this study, we present data on caregivers participating in the psychoeducational intervention combined with the ViveDe 360° immersive videos, which allowed them to experience the same topics covered in the psychoeducational session. At the end of each VR experience, the participants were asked to immediately complete the XRPS scale and the Emotional Attunement Questionnaire, prior to a brief debriefing on the experience and the psychoeducational themes.

At the beginning of the intervention, this group of caregivers was provided with immersive VR fruition equipment (a V2 cardboard box and a stereo headset) and trained in the use of ViveDe at home. In light of the potential for virtual reality (VR) to induce widespread discomfort or feelings of nausea (referred to as cybersickness) in some individuals and the possibility that one’s experience in VR may be influenced by one’s familiarity with the use of such an interactive tool, a special training session was planned for all participants to utilize an ad hoc environment developed in ViveDe for this purpose. Despite the absence of evident limitations in the capabilities of interaction and dizziness phenomena in previous usage sessions of ViveDe’s 360° usable videos in VR, difficulties and resistance to using VR were monitored at this stage and discussed with a psychologist, who was also an experienced VR researcher, with the objective of minimizing the impact of the technology on the sense of presence. This involved addressing perceptual difficulties or awkwardness in interacting with the VR environment, as well as instances of cybersickness. If the difficulty in VR use persisted, the participant could attend the psychoeducational sessions but would be excluded from this study.

## 3. Results

Table 1 summarises the descriptive characteristics of the 36 caregivers and the cared-for person with dementia. Descriptive data were reported as mean (M) supplemented by the standard deviation (SD) or as the number of participants (N) with the percentage in parenthesis.

An initial analysis of the six proposed 360° videos was conducted to understand whether the different VR experiences proposed by ViveDe produced different levels of presence or emotional attunement in the participants. Table 2 describes the mean and median scores for each video. None of the video suggestions appeared to have had a significantly different sense of presence or emotional attunement compared to the others.

Structural equation modeling using R software (version 4.3.2) was used to test for a possible relationship between caregivers’ individual characteristics and ViveDe’s sense of presence and emotional attunement.

Thus, the observed variables of anxiety, depression, caregiver burden, interpersonal reaction, attitude, and emotional reactivity were associated with each other, creating latent variables and delineating a graphical model (see Figure 1 for details).

The empathic concern (EC), perspective taking (PT), personal distress (PD), and fantasy (FS) subscales were considered separately in the model, as the IRI appears to provide a multidimensional measure of the empathy construct [35,36].

As shown, the structural model consists of two latent variables (distress and empathy). Distress was determined by caregiver burden, anxiety, depression, and the IRI personal distress (PD) subscale. Empathy was determined by the empathic concern and fantasy IRI subscales. There was a lack of loading of the IRI perspective-taking (PT) subscale in terms of both distress and empathy factors.

The model showed a good fit (X(42) = 47.313 (42); *p* = 0.265), with a CFI of 0.963 (normal ≥ 0.9), a TFI of 0.929 (normal ≥ 0.9), an RSEA of 0.059 (normal ≤ 0.008), and an SRMR of 0.074 (normal ≤ 0.08) [37]. Composite reliability (CR) and average variance extracted (AVE) indexes were reported to assess the reliability and validity of the latent variables. In particular, AVE (distress = 0.667 and empathy = 0.685) and CF (distress = 0.865 and empathy = 0.788) were above the suggested thresholds (AVE > 0.5 and CF > 0.7; [38]. Figure 2 details the significant associations in the considered model.

The model showed that distress (β = −0.414, *p* = 0.021), empathy (β = 0.523, *p* = 0.010), and age (β = 0.620, *p* = 0.005) significantly predicted the sense of presence. Moreover, distress (β = −0.316, *p* = 0.049), empathy (β = 0.572, *p* = 0.005), and age (β = 0.469, *p* = 0.034) significantly predicted emotional attunement. The model explained 41% of the variance in sense of presence and 48% of the variance in emotional attunement.

## 4. Discussion

As observed from the results, higher age, higher empathy, and less distress were associated with a higher sense of presence and emotional attunement. As for the relationship between older age and changes in the sense of presence experienced in the use of VR tools, this result is not surprising. Research by Dilanchian, Andringa, and Boot [39] compared adults who navigated different VR environments, showing how older adults experienced differing degrees of presence within virtual environments compared to younger ones. In particular, it appears that older people were seen to have a greater sense of presence, especially where this was defined as “involvement”—that is, the attention one gives to the virtual environment and how involved one is with the experience [40]. According to the authors, one reason for this difference might be the lower familiarity older people have with immersive technologies. The “novelty” effect of being able to take a different point of view through VR, in fact, could contribute to generating a greater sense of presence that is less common in young people, since they are more likely to be accustomed to this effect. This conclusion, while interesting, could not be drawn from our experiment, since we had not checked our users’ familiarity with virtual immersive technologies. We can say that most of our users were not at all familiar with VR and had to undergo training with the experimenter before they could enjoy ViveDe at home with cardboard provided for experimentation. This causes us to lean toward the conclusion that the novelty factor was fairly homogeneous for everyone and would not have been such a determining factor in influencing our users’ sense of presence. We can think, however, that having an age closer to that of the family member with Alzheimer’s may have created a greater possibility of identification with what was happening within the virtual experience. This leads us to be able to explore another of the factors that turned out to be significant—namely, empathy. The existing literature does not permit an unequivocal conclusion to be drawn regarding the evolution of different forms of empathy with age, particularly when comparisons are made with younger age groups [41]. The ambiguity of the results is further compounded when attempting to draw comparisons between the empathy developed by older people in virtual reality environments and that of younger people. Indeed, from the results, it was observed that the greater the empathy measured in caregivers, the greater the sense of presence they experienced. In particular, presence seemed to be significant in determining the empathy variable in only some of the subscales of which the IRI is composed—specifically, empathic concern (EC) and fantasy (FS). Regarding empathic concern, our result is congruent with Van Loon and colleagues’ [42] observation that VR experiences can improve other-regarding behaviors, such as empathy. According to Zaki and Ochsner [43], EC is concerned with the ability to share an experience, which can be linked to affective motivation to increase the well-being of another individual. Our caregivers were selected from a much larger number of family members of people with dementia who, during the period in which this study was conducted (August 2022 to May 2024), were referred to Fatebenefratelli Hospital (enrollment ratio = 26.7). This suggests that the people who voluntarily agreed to take part in the psychoeducation groups may have already been largely disposed to wanting to improve the living conditions of their caregivers and thus may have become emotionally attuned to what was happening in the situations experienced in ViveDe.

The fantasy subscale, on the other hand, refers to the tendency to identify with a fictional character, which in the case of ViveDe is quite easy to achieve, since 360° videos involve the first-person perspective (thus, the absence of an avatar to represent the user) and show the environment not as a VR reconstruction but as a real video of everyday situations (e.g., grocery store, bakery, city street, restaurant). This mode breaks the well-known dimension of the theatrical medium [44], which is often attributed to computer-based virtual scenarios and the characters acting in them. In our case, the presence of a 360° video shot in a natural environment with human participants in the scene seems to have promoted a sense of presence, since it did not require any effort on the part of the users to identify with the virtual world and its protagonists.

Finally, the perspective-taking (PT) subscale, which measures cognitive empathy, was not significant in our model. Perhaps this is because the 360° video enjoyed with a cardboard box was not immersive enough to allow users to shift their perspective to someone other than themselves. Alternatively, could it be that the simulation of the symptoms of the disease was perceived by the users as “distant” from what they observed in everyday caregiving, creating a kind of defense/resistance to taking the perspective of the person with dementia in VR? New and more in-depth research is needed to address these questions.

A low level of perceived stress also influenced the sense of presence. While numerous studies have investigated the role of VR in reducing stress, there appears to be no research consistent with our finding of a significant relationship between low levels of caregiver-perceived stress and high levels of presence experienced in VR. We can tentatively conclude that the caregivers who enjoyed ViveDe with less arousal, as determined by their level of distress in real life, may have favored a less “resistant” approach to the VR experience. Indeed, within the 360° videos, the narrative mostly suggests moments of excitement, disorientation, and mental confusion—all situations that caregivers with higher levels of perceived distress may be constantly exposed to while caring for their loved one with dementia. Finding themselves in the same experience in VR might, therefore, have activated a kind of “resistance” to the same experience in VR in those who had already developed high levels of burnout from the daily situation by reducing their level of presence. This conclusion certainly has the limitation of being largely speculative and needs to be investigated in more detail in an ad hoc experimental design.

In conclusion, exposure to 360° video experiences through virtual technologies, as done in ViveDe, seems to have generated a sufficient sense of presence in the family caregivers of people with AD. This effect also seems to have been modulated by individual characteristics, such as susceptibility to certain forms of empathy, perceived stress level, and age proximity to the caregiver. Knowing that these characteristics influence the sense of presence, we can conclude that the virtual experience may be of added value for the psychoeducation of nonprofessional caregivers. Whether this psychoeducational modality can have long-term benefits in the management of patients with dementia will therefore be the subject of future experimental studies.

To date, we can say that our approach has the advantage of being able to be used remotely through low-cost technology. In fact, recent research has shown that more immersive and interactive VR experiences are no more effective in presence than those delivered by cheaper devices, such as cardboard headsets [45]. This finding seems to have been confirmed in our case. Thus, it may soon be possible to develop large-scale psychoeducational interventions that can be used online by caregivers as needed. This would also go a long way in addressing the need often expressed by caregivers that they do not have enough time to educate themselves because they are engaged in caregiving that takes up most of their time away from work.

## 5. Conclusions

At the conclusion of our work, we can say that the use of experiential technologies for a psychoeducation course dedicated to family caregivers of people with dementia has been shown to be effective. Specifically, a sense of first-person involvement in the virtual experience proposed through 360° videos enjoyed through VR cardboard (referred to in the literature as “sense of presence”) was observed. In particular, it was observed how this capacity for experiential immersion in VR worlds was found to differ according to the individual characteristics of the caregivers involved. This result first emphasizes how the sense of presence generated by virtual environments is not a characteristic of VR per se, but can be determined by the characteristics of the environment (the graphical definition, narrative, degree of interactivity, immersion, etc.) in combination with the individual disposition that each user has in enjoying a VR experience. In our case, the age of the participants, their disposition to empathy, and their perceived level of stress seem to have been the main determinants. Since we did not consider other individual and personality factors of the users in this research, we cannot conclude that these are the only determinants of the users’ attributable sense of presence. Future research could go on to investigate how other individual characteristics of the family caregivers involved may influence the sense of presence in virtual environments and provide useful insights into how to better construct psychoeducation programs aimed at caregivers of people with dementia. In addition, a more thorough investigation of individual determinants of sense of presence could guide future research in designing and implementing more meaningful and usable virtual experiences for people with dementia themselves to engage in VR-based treatment pathways.

It can be posited, based on the evidence presented here, that specific individual characteristics are determinants of the sense of presence, and that sense of presence is widely observed in the literature as an essential element for the acquisition of new knowledge through the use of VR. Consequently, it may be suggested that these characteristics be given consideration in the development of psychoeducational interventions designed for the training of caregivers (or potentially for the rehabilitation or treatment of people with dementia pathology). It is therefore necessary to consider not only the adherence of constructed environments to natural reality but also to analyse the characteristics of user interaction with technology and the personal attributes of end users in order to determine the sense of presence and thus the efficacy of the psychoeducational virtual experience.

In conclusion, the introduction of virtual psychoeducation tools may facilitate a more nuanced understanding of the patient’s perspective and potential caregiver contributions. Additionally, integrating innovative elements within a psychoeducation program may enhance user engagement and adherence. This adherence may be attributed to curiosity and enjoyment associated with this novel interactive element, which may serve as a motivating factor for program participation. It is not possible to conclude with certainty that this is the benefit obtained, as no data is available that measures adherence to the program in comparison with a control group. In the near future, we intend to investigate whether the inclusion of virtual reality (VR) in psychoeducation programs affects adherence, compliance, and motivation to follow these programs. This will enable us to understand the impact of this technology on these factors.

The present study in the use of advanced immersive technologies such as virtual reality poses some limitations that have not been evident so far from the research results. One among them is the familiarity that users of a generation considered already adults may have with the use of technology. In the specifics of our research, most of the users involved had had no prior familiarity with the use of immersive technologies. This resulted in some differences within the group of family caregivers with respect to the psychological predisposition that the users themselves might have had in enjoying the virtual reality experience. Based on the indications given by the experimenters in the training sessions on the use of the technology prior to the trial, we can imagine that some effects on the sense of presence may have been determined by greater or lesser confidence in the use of VR. In addition, we can point out that the ViveDe sessions were enjoyed independently by the users in an online psychoeducation course without the presence of an operator who could possibly intervene, even remotely, in facilitating the enjoyment of the experience or in providing guidance to compensate for some technical drawbacks. This is certainly one of the most important limitations of the present study, and it was partly considered by the experimenters to be a qualitative element not analyzed in the results. At the same time, it could also be seen as an element of the strength of the project itself, since having been able to use a technology such as virtual reality in the home through a remote psychoeducation course is in fact one of the points of originality of this research. To date, in fact, from a review of the scientific literature, it does not seem to us that VR has been used in this way in psychoeducational settings for caregivers of people with dementia. In fact, precisely the use of low-cost technology that has also been found to be usable by remote users seems to be a new possibility in this area. New research oriented in this direction should, in our opinion, deepen this opportunity in the coming years by expanding the sample of reference, as well as by taking advantage of the rapid development of virtual immersive technologies that could go on to massively involve a good portion of the population of informal caregivers.

It is evident that this study is constrained by two significant factors: the limited sample size and the absence of a control group. As previously outlined in the methodology section of this study, the research aimed to investigate the efficacy of both conventional and enhanced forms of psychoeducation with virtual reality experiences in a population of informal dementia caregivers. Specifically, we sought to ascertain whether the individual characteristics of the caregivers might have influenced their enjoyment of the VR experience. Consequently, when the population size was not yet large, we conducted this analysis, which we present here. This study can be considered an initial investigation into a novel phenomenon. It is, to our knowledge, the first of its kind in the scientific literature on dementia caregiving. As such, it paves the way for future research that will delve deeper into individual determinations of sense of presence, potentially including a control group to gain a more nuanced understanding of the phenomenon.

The use of virtual reality in the psychoeducation of individuals who are the primary caregivers of people with dementia is currently in its infancy. However, there is a growing body of research that suggests this may be an effective method. Therefore, we can conclude that, although more research is needed, virtual reality has the potential to be a valuable tool for this purpose. The acquisition of skills and knowledge regarding dementia pathology through a first-person perspective represents a promising avenue for exploration. However, before definitive conclusions can be drawn about the effectiveness of virtual reality for training family caregivers in the care of individuals with chronic neurodegenerative conditions such as dementia, this type of experimentation must still undergo a significant expansion of its knowledge base.

## Figures and Tables

**Figure 1 brainsci-14-00852-f001:**
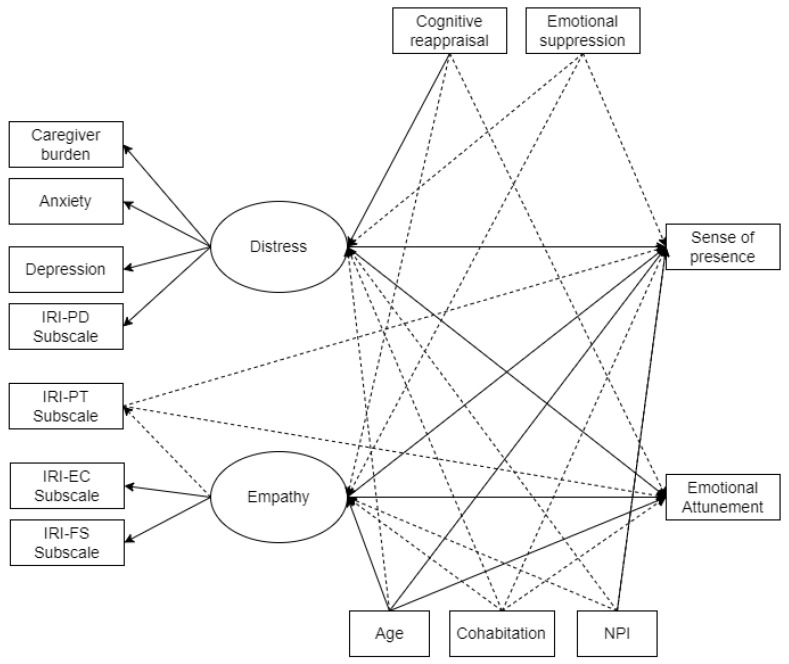
Complete structural equation model with all tested pathways. Notes: Circle: latent variables; rectangle: observed variables; single arrows: regression relation; dotted arrows: non-significant pathways (*p* > 0.05). IRI-PD Subscale = Interpersonal Reactivity Index–Personal Distress Subscale; IRI-PT Subscale = Interpersonal Reactivity Index–Perspective Taking Subscale; IRI-EC Subscale = Interpersonal Reactivity Index–Empathic Concern Subscale; IRI-FS Subscale = Interpersonal Reactivity Index–Fantasy Subscale.

**Figure 2 brainsci-14-00852-f002:**
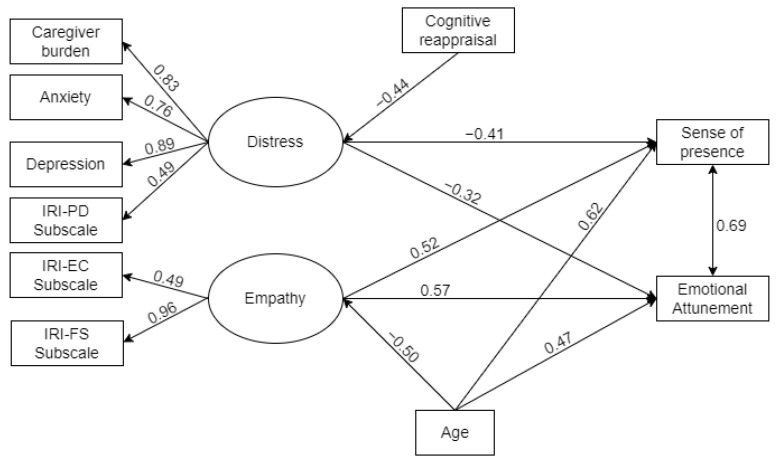
Structural equation model analysis: standardized regression weights of variables in relation to sense of presence and emotional attunement. Note: Only significant pathways (*p* < 0.05) are shown in the figure. Notes: Circle: latent variables; rectangle: observed variables; single arrows: regression relation. IRI-PD Subscale = Interpersonal Reactivity Index–Personal Distress Subscale; IRI-PT Subscale = Interpersonal Reactivity Index–Perspective Taking Subscale; IRI-EC Subscale = Interpersonal Reactivity Index–Empathic Concern Subscale; IRI-FS Subscale = Interpersonal Reactivity Index–Fantasy Subscale.

**Table 1 brainsci-14-00852-t001:** Characteristics of caregivers and PwD (N = 36).

Features of Caregivers of PwD	M ± SD or N (%)
Age	53.81 ± 10.81
Sex-*Female*	29 (80.6)
Education level	
*None*	0
*Primary school*	0
*Middle school*	4 (11.1)
*High school*	18 (50.0)
*University*	13 (36.1)
*Post-lauream*	1 (2.8)
Relationship with PwD	
*Partner*	9 (25.0)
*Child*	25 (69.4)
*Other*	2 (5.6)
Cohabiting	
*Yes*	15 (36%)
*No*	21 (58%)
Caregiving time	
*<6 months*	3 (8.3)
*6 months–1 year*	4 (11.1)
*1–2 years*	15 (41.7)
*2–3 years*	6 (16.7)
*3–4 years*	1 (2.8)
*4–5 years*	3 (8.3)
*>5 years*	4 (11.1)
Caregiving hours/day	
*2–4 h*	17 (47.2)
*4–6 h*	4 (11.1)
*6–8 h*	3 (2.8)
*8–10 h*	0
*10–12 h*	1 (2.8)
*>12 h*	5 (11.1)
*24 h*	6 (25.0)
Features of PwD	
*Age PwD*	77.08 ± 7.19
*Duration disease PwD*	2.19 ± 1.69
*NPI PwD*	9.39 ± 7.56

PwD = person with dementia; NPI = Neuropsychological Inventory.

**Table 2 brainsci-14-00852-t002:** XRPS presence scale and emotional attunement questionnaire mean and median scores for each VR experience proposed by ViveDe’s 360° immersive videos (N = 36).

Video	Topic	XRPS		Emotional Attunement	
		M (SD)	Med (IQR)	M (SD)	Med (IQR)
1	Experiencing diagnosis and poor communication	2.83 (0.83)	3 (2–4)	3.69 (0.86)	3.5 (3–4)
2	Experiencing difficulty in cooking	2.57 (0.90)	2.5 (2–3)	3.25 (0.88)	3 (2–4)
3	Experiencing getting lost	2.59 (1.06)	2 (1–4)	3.31 (0.94)	3 (2–4)
4	Experiencing difficulty in shopping	2.86 (1.02)	3 (2–4)	3.48 (0.80)	3 (2–4)
5	Experiencing not knowing one’s home	2.63 (1.04)	2 (2–3)	3.13 (0.93)	3 (2–4)
6	Experiencing difficulty in eating with the family	2.85 (0.97)	3 (2–4)	3.61 (0.80)	4 (3–4)

XRPS: Extended Reality Presence Scale.

## Data Availability

Data supporting reported results can be found on publicly archived dataset generated during the study available at https://clinicaltrials.gov/ (number NCT05780476).

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
