# Peer review of "Virtual Reality-Based Psychoeducation for Dementia Caregivers: The Link between Caregivers’ Characteristics and Their Sense of Presence"

_brainsci, 2024, doi:10.3390/brainsci14090852_

Round 1

Reviewer 1 Report

Comments and Suggestions for Authors

Dear Authors,

Your research on experiencing the perspective of dementia patients through virtual reality is both innovative and crucial in addressing the challenges of dementia care. I believe this study has the potential to significantly contribute to the field. To further enhance its value, I would like to offer the following suggestions:

Introduction:

Elaborate on existing caregiver-focused psychoeducational programs, detailing their types and limitations. This would strengthen the rationale for implementing innovative approaches such as VR.

Include more citations of previous studies exploring the relationship between sense of presence and behavioral changes in caregivers. This would reinforce the purpose and significance of your study.

While the application of VR is innovative, a discussion of its potential limitations would provide a more balanced perspective.

Methods:

Address the small sample size and lack of a control group in the study limitations section.

Provide information on the reliability and validity of each measurement tool used.

Include details about participants' familiarity with VR technology and any pre-study experience provided (e.g., ViveDe 360 familiarization sessions). Additionally, discuss any considerations or countermeasures for potential VR-induced side effects (e.g., dizziness, discomfort).

Results:

Offer a more detailed interpretation of the unexpected finding that older participants reported higher levels of presence and emotional attunement in the VR experience. Present possible explanations for this result.

Discussion:

Expand on the relationship between age and sense of presence beyond the 'novelty effect'. Consider alternative explanations, such as the possibility of deeper empathic abilities in older caregivers.

Include suggestions for the design of VR-based caregiver education programs based on your findings.

Highlight the advantages of this VR-based approach compared to traditional non-VR psychoeducational programs.

Addressing these points would further elevate the academic rigor and practical implications of your valuable research. I look forward to seeing the impact of this study on dementia care practices.

Author Response

Thank you very much for taking the time to review this manuscript. Please find the detailed responses below and the corresponding revisions/corrections highlighted/in green in the re-submitted files.

Comment 1: "Elaborate on existing caregiver-focused psychoeducational programs, detailing their types and limitations. This would strengthen the rationale for implementing innovative approaches such as VR."

Response 1: Thank you for your comment that allowed us to introduce a recent meta-analysis by Gosh and collaborators included in the bibliography and on page 2 (lines 68-77) defining the still unclear effect of psychoeducation interventions on caregivers of people with dementia. We hope that this change may go in the direction suggested by the reviewer and better introduce the approach based on experiential technologies such as VR

Comment 2: "Include more citations of previous studies exploring the relationship between sense of presence and behavioral changes in caregivers. This would reinforce the purpose and significance of your study."

Response 2: In lines 114-117 we emphasized the role of the human and contextual factor (not only technological)in determining the sense of presence. In addition, we linked it with a brief reflection on the generalization of knowledge gained in VR settings to the natural world, having no evidence of substantial scientific literature to support the specific effect of the sense of presence experienced in VR on dementia caregivers' behaviors.

Comment 3: "While the application of VR is innovative, a discussion of its potential limitations would provide a more balanced perspective."

Response 3: We have added the limitations of VR as suggested (lines 136-146). By indulging this valuable comment actually, the introduction looks better balanced. We would like to sincerely thank the reviewer for this opportunity that will make our manuscript more interesting.

Comment 4: "Address the small sample size and lack of a control group in the study limitations section"

Response 4: Thank you again for this comment that allows us to enrich a paragraph on the limitations of the study that was also suggested to us by another reviewer. You will find this specific part in green in the new text (lines 462-483) within the conclusions

Comment 5: "Provide information on the reliability and validity of each measurement tool used"

Response 5: All of the scales we introduced in the research have a validity study behind them, and we have reported the citations. However, we have specified this point (lines 186-188) to better clarify the doubt that the reviewer. We thank you for this comment that allowed us to clarify the construct validity of the evaluation tools used

Comment 6: "Include details about participants' familiarity with VR technology and any pre-study experience provided (e.g., ViveDe 360 familiarization sessions). Additionally, discuss any considerations or countermeasures for potential VR-induced side effects (e.g., dizziness, discomfort)."

Response 6: Thank you for the comment that provide us the possibility to stress the role of training in VR to avoid individual differences in familiarity with interactive technologies and cybersickness disposition (lines 261-271)

Comments 7- 8 "Offer a more detailed interpretation of the unexpected finding that older participants reported higher levels of presence and emotional attunement in the VR experience. Present possible explanations for this result." "Expand on the relationship between age and sense of presence beyond the 'novelty effect'. Consider alternative explanations, such as the possibility of deeper empathic abilities in older caregivers."

Response 7-8: We have addressed this topic in 334-355 lines. To stress this data we added a small part about empathy modification with age (356-361). Unfortunately, the scientific literature doesn't seem to support us in deeply discussing this result. 

Comment 9: "Include suggestions for the design of VR-based caregiver education programs based on your findings."

Response 9: We regard this point as very important and thank the reviewer for providing the opportunity to emphasize it (lines 451-460). Very often in the use of VR, more emphasis is placed on the technological part than on the human part, and it is not our intention to go in this unambiguous direction

Comment 10: "Highlight the advantages of this VR-based approach compared to traditional non-VR psychoeducational programs"

Response 10: We have briefly included this part as suggested (lines 461-471). Unfortunately, from the present research we do not have sufficient elements to be able to compare VR-based psychoeducation with a traditional program

Comment 11: "Addressing these points would further elevate the academic rigor and practical implications of your valuable research. I look forward to seeing the impact of this study on dementia care practices."

Response 11: Thank you very much. We considered these comments really valuable to improve our contribution. We hope in the near future to encounter such timely and constructive reviews again

Reviewer 2 Report

Comments and Suggestions for Authors

1-     The research offers a new perspective to increase the effectiveness of interventions for the problems experienced by primary caregivers of individuals diagnosed with dementia. Therefore, I believe that it is an important study. I have a small suggestion:

In page 2, line 72-74, the sentence begin “The objective of this investigation is to identify the resources a caregiver can rely on……..” Is this the aim of your study or does it belong to Lee et al.'s study? I suggest you clarify this.I suggest you clarify this.

Author Response

Comment 1: "The research offers a new perspective to increase the effectiveness of interventions for the problems experienced by primary caregivers of individuals diagnosed with dementia. Therefore, I believe that it is an important study."

Response 1: We would like to thank the reviewer for the encouraging comment and for appreciating our research. We hope that the psychoeducational approach we presented can contribute to the well-being of people with dementia and their family caregivers.

Comment 2: "In page 2, line 72-74, the sentence begin “The objective of this investigation is to identify the resources a caregiver can rely on……..” Is this the aim of your study or does it belong to Lee et al.'s study? I suggest you clarify this."

Response 2: Thank you for this comment that allows us to disambiguate the indicated sentence that indeed appears ambiguous upon closer rereading. In line 72 is highlighted in blue the change that is "The objective of the investigation we present here ...." It is the objective of our research, the reference to Lee in the next line is to confirm the importance of perspective-taking. We hope that posed in this way the sentence will be clearer to the reader.

Reviewer 3 Report

Comments and Suggestions for Authors

This is an interesting article about psycoeducation of demented patients’ caregivers using virtual reality. The article is generally well structured.

Introduction section sets the background of the current study as it describes dementia, the modification of family roles after diagnosis, the need of caregivers for psycoeducation and the potential role of virtual reality.

Methodology section is descriptive enough, referring to the participants of the study, the measurements performed and the materials and methods that were implemented during this study.

Results are quite interesting. They are also well depicted in tables and figures, making it easier for the potential readers to understand.

In discussion, authors try to summarize their findings and critically discuss them based on literature data. I think that they should add a paragraph, describing the main limitations of their study.

Finally, I think that the conclusion should be written in a separate paragraph, proposing some specific targets for future studies too.

References are relative to the subject. Perhaps the authors could add some more recent references.

Author Response

Commnets 1-4: "

This is an interesting article about psycoeducation of demented patients’ caregivers using virtual reality. The article is generally well structured.

Introduction section sets the background of the current study as it describes dementia, the modification of family roles after diagnosis, the need of caregivers for psycoeducation and the potential role of virtual reality.

Methodology section is descriptive enough, referring to the participants of the study, the measurements performed and the materials and methods that were implemented during this study.

Results are quite interesting. They are also well depicted in tables and figures, making it easier for the potential readers to understand."

Response : We would like to thank the reviewer for valuable comments supporting our research contribution. We would also like to thank him or her for suggestions that make our manuscript clearer to read and improved in terms of argumentation. All changes to this new version of our article that follow the reviewer's suggestions are indicated in orange 

Comment 5 and 6: "In discussion, authors try to summarize their findings and critically discuss them based on literature data. I think that they should add a paragraph, describing the main limitations of their study."

Response 5 and 6: Thank you for pointing this out. We concord with the need to include a conclusion paragraph in which to discuss the limitations of the study. We have added this part as suggested mentioning study limitations and next future research  (Lines 387-436)

Comment 7: "References are relative to the subject. Perhaps the authors could add some more recent references."

Response 7: We agree with this comment. Therefore, for how concerns literature on presence we have included literature that, in the peak period of VR development (late 1990s of last century and early 10s of this century) investigated the determinants of this experience with immersive technologies. For this reason, the citations seem to be out of date. In addition, this less technology-oriented approach that considers the human factor is characteristic of this scientific period (perhaps attributable to a lower level of technological development that valued the human factor). We hope that the kind reviewer understands the reasons for this choice and can accept our motivation for privileging such citations.

Reviewer 4 Report

Comments and Suggestions for Authors

The efforts of many researchers who have conducted research on interesting topics will be good news for dementia patients and their families. However, for the following reasons, this paper needs to be revised. 1. It is necessary to confirm whether the purpose of this study is appropriate for the purpose of this journal. The purpose of this journal is as follows: 'The aim is to encourage scientists to publish their experimental and theoretical results in as much detail as is required to fully convey the information.' 2. It is questionable whether the burden of caregivers who play the role of dementia patients' family members can be alleviated through such online education. Especially, the form of [education] requires education based on the neurological mechanisms of dementia, such as cognitive understanding, learning of information, and application of skills. Therefore, the word "psychoeducation" in the title of this study is unclear. 3. Currently, hospitals approach the treatment and care of AD, dementia, and MCI differently. However, the study title mentions dementia, and the subjects cared for by caregivers are at the MCI level, and a standardized evaluation that accurately diagnoses AD other than MMSE is not presented. Therefore, accurate understanding of the diagnosis is necessary. 4. It is necessary to provide a detailed explanation of the theoretical framework on which the VR contents are based. 5. The age of the research subjects is 43-63 years old, and the cognitive level of MMSE is 18-24 points, and they care for AD at the same cognitive level. It is unclear whether the research subjects have the same cognitive level as AD that caregivers care for and whether the baseline is sufficient to understand this VR education. 6. It is necessary to confirm whether validity testing by experts in degenerative neurological disorders, media professionals, and elderly experts was conducted to apply the developed contents to clinical practice. 7. The composition and arrangement of the content in the main body are not appropriate, and the core contents that are creative, academic, and clinical need to be summarized concisely. However, the researchers' interdisciplinary approach to experiencing the cognitive and socio-emotional level of caregivers' AD through VR is excellent, so the content should be appropriately revised for the purpose of this study and continued efforts should be made to publish it in an appropriate scope of the journal.

Author Response

Comment 1 : "It is necessary to confirm whether the purpose of this study is appropriate for the purpose of this journal. The purpose of this journal is as follows: 'The aim is to encourage scientists to publish their experimental and theoretical results in as much detail as is required to fully convey the information."

Response 1: We would like to express our sincerest gratitude to the reviewer for this invaluable clarification. We consider it of paramount importance to ensure that our submissions align with the aims and objectives of the scientific journals to which we submit our work. This submission is submitted in response to a request from Brain Science to consider this venue as one of the possible ones for evaluating our research. Consequently, upon accepting the invitation, we submitted an abstract to the editor for his approval of our research objectives and suitability for the journal. We do not believe that this preliminary step guarantees us publication in Brain Science. Nevertheless, we also consider it may have prevented some confusion regarding the appropriate alignment of our proposal with the aims of this journal. In the event of any misunderstanding regarding this type of matching, we extend our sincerest apologies to the editor and all the reviewers who participated in the review process, offering our gratitude for their constructive comments and time. 

Comment 2: " It is questionable whether the burden of caregivers who play the role of dementia patients' family members can be alleviated through such online education. Especially, the form of [education] requires education based on the neurological mechanisms of dementia, such as cognitive understanding, learning of information, and application of skills. Therefore, the word "psychoeducation" in the title of this study is unclear."

Response 2: We are grateful to the thoughtful reviewer for their feedback. However, we believe that an understanding of the pathophysiological mechanisms of dementia, while valuable, does not fully equip caregivers with the necessary tools to care for individuals with dementia effectively. In each instance, we have included a definition of psychoeducational according to Yu et al. 2023 (lines 77-71 in red) and a reference to literature indicating the efficacy of psychoeducation programs in reducing caregiver distress, albeit to a limited extent (lines 68-76 in green as the other modification relatives to reviewer 1). 

Comment 3: "Currently, hospitals approach the treatment and care of AD, dementia, and MCI differently. However, the study title mentions dementia, and the subjects cared for by caregivers are at the MCI level, and a standardized evaluation that accurately diagnoses AD other than MMSE is not presented. Therefore, accurate understanding of the diagnosis is necessary."

Response 3: The diagnosis of Alzheimer's disease, which is a prerequisite for inclusion in this research, was conducted by neurologists and neuropsychologists at Fatebenefratelli Hospital in accordance with the clinical standards established by the scientific community for the assessment of this disease. This report details the MMSE, which was used as the baseline data for the patient's cognitive impairment. However, all diagnostic investigations were conducted in the clinic to assess whether the patient was suffering from Alzheimer's-type dementia. 

Comment 4: "It is necessary to provide a detailed explanation of the theoretical framework on which the VR contents are based."

Response 4: Thank you for this comment that allows us to elaborate on how the ViveDe tool was implemented. Referring to our previous publication (Morganti et al. 2020), we have delved into the design features that led to the development of this virtual reality program (lines 220-224 in red)

Comment 5: "The age of the research subjects is 43-63 years old, and the cognitive level of MMSE is 18-24 points, and they care for AD at the same cognitive level. It is unclear whether the research subjects have the same cognitive level as AD that caregivers care for and whether the baseline is sufficient to understand this VR education."

Response 5: We would like to thank the reader for bringing an apparent inconsistency in the paper to our attention. This was caused by a typographical error and has now been corrected. The ages of the caregivers and people with dementia have been specified, as has the MMSE value, which now refers to people with dementia rather than their caregivers. We hope that the paper is now easier to read. 

Comment 6: " It is necessary to confirm whether validity testing by experts in degenerative neurological disorders, media professionals, and elderly experts was conducted to apply the developed contents to clinical practice."

Response 6: The assessment of individuals with dementia and the psychoeducation of their caregivers were conducted by physicians, psychologists, and psychotherapists who are duly registered with professional registries and possess the requisite qualifications to engage in such practice.  We have already specified it in the manuscript. For the reviewer's convenience, these sections have been highlighted in red

Comment 7: "The composition and arrangement of the content in the main body are not appropriate, and the core contents that are creative, academic, and clinical need to be summarized concisely. However, the researchers' interdisciplinary approach to experiencing the cognitive and socio-emotional level of caregivers' AD through VR is excellent, so the content should be appropriately revised for the purpose of this study and continued efforts should be made to publish it in an appropriate scope of the journal."

Response 7: We are grateful to the reviewer for their observation regarding the interdisciplinary nature of our research group. In regard to the suitability of our submission in relation to the aims of the journal, we respectfully defer to the editor's decision, citing the previously addressed comment as a reference.